# Effects of Live Music Therapy on Autonomic Stability in Preterm Infants: A Cluster-Randomized Controlled Trial

**DOI:** 10.3390/children8111077

**Published:** 2021-11-22

**Authors:** Dana Yakobson, Christian Gold, Bolette Daniels Beck, Cochavit Elefant, Sofia Bauer-Rusek, Shmuel Arnon

**Affiliations:** 1Music Therapy Department, Institute for Communication and Psychology, Aalborg University, 9000 Aalborg, Denmark; bolette@hum.aau.dk; 2Neonatal Department, Meir Medical Center, Kfar Saba 44281, Israel; Bauers@clalit.org.il (S.B.-R.); shmuelar@clalit.org.il (S.A.); 3NORCE Norwegian Research Centre AS, 5008 Bergen, Norway; chgo@norceresearch.no; 4Department of Clinical and Health Psychology, University of Vienna, 1010 Vienna, Austria; 5School for Creative Arts Therapies, University of Haifa, Haifa 3498838, Israel; celefant@univ.haifa.ac.il; 6Sackler Faculty of Medicine, Tel Aviv University, Tel Aviv 6997801, Israel

**Keywords:** music therapy, preterm infants, family-centered care, autonomic stability, heart rate variability

## Abstract

Unbuffered stress levels may negatively influence preterm-infants’ autonomic nervous system (ANS) maturation, thus affecting neurobehavior and psycho-emotional development. Music therapy (MT) is an evidence-based treatment modality in neonatal care. When coupled with skin-to-skin care (SSC), it may reduce stress responses in both preterm infants and their parents and enhance family-centered care. Accordingly, we aimed to compare the effects of combined MT and SSC and SSC alone on ANS stabilization in preterm infants. In a single-center, cluster-randomized trial design, ten two-month time-clusters were randomized to either combined MT and SSC or SSC alone. Families of preterm infants were offered two sessions of the allocated condition in the NICU, and a three-month follow up session at home. The primary outcome variable was stabilization of the ANS, defined by change in the high frequency (HF) power of heart rate variability (HRV) during the second session. Secondary outcomes included other HRV measures, parent–infant attachment, and parental anxiety at each session. Sixty-eight families were included. MT combined with SSC improved infants’ ANS stability, as indicated by a greater increase in HF power during MT compared to SSC alone (mean difference 5.19 m^2^/Hz, SE = 1.27, *p* < 0.001) (95% confidence interval 0.87 to 2.05). Most secondary outcomes were not significantly different between the study groups. MT contributes to preterm-infants’ autonomic stability, thus laying an important foundation for neuro-behavioral and psycho-emotional development. Studies evaluating longer-term effects of MT on preterm infants’ development are warranted.

## 1. Introduction

Neonatal stress exposure is common treating premature infants. The intense medical care, stimulating neonatal intensive care unit (NICU) environment, and parental separation are all major contributors to preterm infants’ stress experiences [1], leading to over stimulation during a critical period of neurodevelopment [2]. Preterm infants’ autonomic nervous system (ANS) is highly vulnerable due to immaturity and neuroplasticity [3], and is chronically triggered with repeated fight-or-flight reactions to the external stressors. Disruption of the ANS maturation may lead to an allostatic load [4], diminish their ability to cope, and harm synaptogenesis development and neurodevelopmental outcomes [5,6]. Simultaneously, the preterm-birth and NICU conditions may lead parents to experience posttraumatic stress, anxiety, and depression [7], which affect parent–infant bonding processes and further impact infants’ developmental outcomes [1].

Combined skin-to-skin care (SSC) and music therapy (MT) may promote autonomic and physiologic stabilization in preterm infants [8], reduce stress in parents [9], and further contribute to their ability to handle the stressful effects of the NICU conditions. As SSC is the main form of physical bonding between preterm infants and their parents in the NICU [10], transmitting the parental voice or music listening during SCC may consider multi-sensory precautions and promote family-centered care [11]. This type of multimodal intervention may have an additive beneficial affect for preterm infants [12]. However, combined MT and SSC have been investigated in only few, under-powered studies [13,14,15,16]. One study demonstrated that maternal singing during SSC significantly improved preterm-infants’ autonomic stability and maternal anxiety compared to SSC alone [17], although without giving therapeutic support to the parent–infant dyad. Forming intimacy and meaningful interactions in the high-tech medical NICU environment may be challenging [18,19]. Support from a professional NICU music therapist during SSC may enhance these early bonding experiences and improve infants’ ANS regulation by providing an informed sensitivity to infants’ communication cues, parental needs, and modifying their physical sound environment [20].

There is a paucity of neonatal MT trials examining sensitive physiological markers such as heart rate variability [21], which is recognized as an important indicator of ANS maturation [22]. Furthermore, a recent meta-analysis [9] and two systematic reviews [21,23] highlighted the lack of rigor in MT trials within a family-centered approach facilitated by a certified NICU music therapist, and long-term outcomes. Accordingly, we aimed to examine the effects of a family-centered MT intervention during SSC. We conducted a cluster-randomized controlled trial (cRCT), to compare the effects of combined family-centered MT and SSC and SSC alone, on preterm-infants’ autonomic stability and parental wellbeing during the NICU admission and at three-month follow-up. Our hypothesis was that combined MT and SSC, as compared to SSC alone, would enhance better ANS regulation, decrease parental stress, and increase parent-to-infant attachment.

## 2. Materials and Methods

### 2.1. Design

In this single center cRCT with two parallel arms, 10 time-clusters of two months each were allocated to either combined MT and SSC or SSC alone. Constrained randomization of pairwise matching clusters was conducted by an external researcher with no contact with participants (C.G.). The time-clusters design was chosen to prevent the high potential of contamination bias between groups due to the open bay setting of the participating NICU. Each time cluster lasted for two months, corresponding to the average length of stay for infants in the participating NICU, namely, six weeks. In each time-cluster, all parent–infant dyads (mother–infant or father–infant, separately) participated in two sessions of the assigned intervention, and in a three-month follow-up session at home.

### 2.2. Setting and Procedures

Sessions lasted 30–45 min. Recommended sound volumes of an hourly Leq of 50 dB [24] were monitored using a sound analyzer (Extech SL130, Extech Instruments, Nashua, NH, USA). Normothermia in infants was controlled by a temperature probe for continuous monitoring. The study was conducted in the NICU at Meir Medical Center in Kfar Saba, Israel, a 24-bed, level III, tertiary NICU with Newborn Individualized Developmental Care and Assessment Program (NIDCAP) certification. Screening, recruitment, and informed consent were conducted by the senior neonatologist (S.A.) and a NIDCAP trainer. The trial was prospectively registered in ClinicalTrials.gov, NCT03023267, and the full trial protocol was published [25].

### 2.3. Participants

All clinically stable preterm infants with gestational age (GA) < 36 weeks and confirmed hearing based on otoacoustic emissions tests were eligible. The hearing test was conducted prior to study entry by a specialized assessor at the NICU environment. Clinical stability was assessed by the senior neonatologist and included no mechanical ventilation or acute illness. Severity of prematurity was categorized according to the neonatal medical index [26]. All parents with sufficient understanding of the study language were eligible. Infants were excluded due to medications acting on the central nervous system, any respiratory distress with oxygen support, intraventricular hemorrhage stage ≥ 3, periventricular leukomalacia, active apnea episodes who require medical intervention, or an estimated hospitalization of less than ten days. Additionally, pre-study data were collected from medical charts and included bronchopulmonary dysplasia, necrotizing enterocolitis, respiratory distress syndrome, retinopathy of prematurity, and sepsis episodes. Parents with language or cognitive difficulty preventing compliance with study procedures were excluded.

### 2.4. Clusters Randomization and Recruitment

Between May 2017 and September 2018, 10 clusters were randomized. This comprised five in each arm, including 68 eligible families who consented and were enrolled. The intraclass coefficient (ICC) was 5%, indicating a moderate effect of the clustering. Baseline characteristics of 68 infants (MT: *N* = 37, SSC: *N* = 31), and 79 parents (MT: *N* = 42, SSC: *N* = 37) were similar across the two arms (Table 1), indicating that the cluster randomization achieved baseline balance. Infants’ mean gestational age was 30.56 ± 2.66 and 31.06 ± 2.92 weeks (in MT and SSC groups, respectively). In both groups, 75% of the dyads completed at least three sessions. In most families only one parent participated, usually the mother (*N* = 47/68 infants, 72%).

### 2.5. Interventions

Experimental group: Family-centered MT during SSC (hereafter, referred to as MT). The intervention was based on the “First Sounds: Rhythm, Breath and Lullaby” (RBL) model [27], and facilitated by a certified RBL-music therapist (D.Y.). After SSC placement, parents were instructed to entrain their breathing patterns to their infants’ respiratory rates, and by doing so, to gradually stabilize both. The music therapist accompanied them using the Remo Ocean disk, an instrument especially designed to promote breathing and relaxation by resembling the intrauterine sound environment [28]. Parents were then guided to hum in repeated, simple, melodic patterns. The humming gradually developed to singing two to three songs of the parents’ choice, adapted to a lullaby rhythm (i.e., “song of kin” [29]). These songs were accompanied by guitar music according to parents’ preferences. The intervention guidelines [25] allowed for flexibility to address alternating parental or infants’ needs. Some parents asked for vocal or instrumental support prior to taking the active role and singing or preferred to receptively listen to their songs played live by the music therapist. In any case, they were encouraged to softly talk to their infant, tell a story or share their emotional experiences.

Control group: SSC alone (hereafter referred to as SSC), as described previously [30]. SSC practice is considered standard care in the participating NICU. Parents were guided to act as they usually would during SSC time, with the restriction of singing. No therapeutic conversation was offered. Infants were monitored through all sessions by a trained research assistant. The intervention was paused at any sign of stress and was treated by the infants’ nurse. Further details of both interventions are described in the protocol [25].

### 2.6. Outcomes

Primary outcome: Stabilization of the autonomic nervous system (ANS) and reduced stress in preterm infants, as indicated by a change in the high frequency (HF) power of their heart rate variability (HRV). We examined the HF change during the second session (i.e., from the first to the last part of the session). The first intervention session was considered more preparatory, and so we decided to focus on the second session assuming by then parents would have been familiarized with the process. HRV is the spontaneous beat-to-beat variations in heart rate. These variations originate from the autonomic nervous system (ANS) and provide information about its parasympathetic and sympathetic branches [31]. It is measured through spectral power analysis of its high and low frequency (HF, LF) domains. The HF band at 0.15–1.8 Hz is considered to reflect vagal efficiency in the parasympathetic system, an indicator of rest-and-digest reactions. The LF band at <0.15 Hz may indicate sympathetic activity, which occurs during the fight-or-flight reaction, or parasympathetic withdrawal [32]. Accordingly, the LF/HF ratio may reflect the sympatho-vagal balance, a relationship between relaxation and stress [33]. High values of HF and low values of LF and LF/HF ratio are considered signs of ANS stability and low stress [17]. HRV analysis was conducted using the electrocardiogram (ECG) analogue signal from the cardiorespiratory monitor (Philips, Agilent monitors, Irvine, CA, USA) [34]. The monitor records infants’ heart rate, respiratory rate, and oxygen saturation, as part of their routine care. The ECG analogue signals were recorded continuously during all sessions, from 10 min before until 10 min after the end of the session. These records were fed by an external neonatologist into an HRV software program (ANSR1000 system Ansar, Inc., Philadelphia, PA, USA) and converted to digital values, reflecting cyclic changes. The software’s algorithm eliminates movements and artifacts and transforms the data into a waveform across a spectrum of frequencies measured in Hz using the geometric mean.

Secondary outcomes—preterm infants: Change in LF power and LF/HF ratio during the second session, and mean HF, LF, and LF/HF ratio across all three sessions were considered as secondary outcomes. Additional infant outcomes (fine-grained within-session change) will be reported separately.

Parental outcomes: Parent-to-infant attachment level was measured with the maternal postnatal attachment scale (MPAS) [35]. This validated, 19-item self-report questionnaire is scored on a five-point Likert scale, describing parents’ behaviors, attitudes, and feelings towards their baby. Total scores range from 19 to 95, and higher scores represent higher attachment. The MPAS was obtained at study entry, after one month, and in the three-months follow-up. State anxiety was measured using the State–Trait Anxiety Inventory (STAI) [36]. It includes 20 self-report statements evaluating current anxiety symptoms, scored on a scale from 1 (not at all) to 4 (very much). Scores range from 20 to 80, where higher scores indicate higher degree of anxiety. The STAI was administered before and after each session.

### 2.7. Power Calculation

The minimal clinically important difference for HF power in infants is unknown. Accordingly, the power calculation for the primary outcome was informed by a previous study [17], which found a mean value of 16.8 ms^2^/Hz for HF power during combined maternal singing and SSC, as compared to 10.5 ms^2^/Hz during SSC alone (effect size *d* = 1.44). We hypothesized that in the current study, HF values may increase more, up to 20 ms^2^/Hz, due to the family-centered approach. The study was powered to detect a large effect (*d* = 0.80). Considering the clustered nature of the data, intraclass coefficient (ICC) = 0.01, and 20% attrition, the total required sample size was 58 [25].

### 2.8. Data Analysis

Analysis of all outcomes was based on the intention-to-treat principle using linear mixed effects (LME) models. Descriptive methods were used for baseline characteristics. Normality of continuous data was assessed graphically. Comparison of effects between groups included bivariate tests for continuous variables and Pearson Chi-squared tests for categorical variables. A two-sided, 5% significance level was set for the primary outcome, and a 1% level for secondary outcomes. LME models examined changes over time or effects during a session for all repeated-measures variables. Random effects included time-cluster and infant (i.e., if more than one parent participated). Repeated effects included the time points and type of therapy. Covariates’ examination included infants’ sex, GA, postmenstrual age at study entry, birthweight, and neonatal medical index grade. Analyses were conducted using SPSS for Windows, version 14 (IBM Corp., Armonk, NY, USA) and R version 3.5.0 [37].

## 3. Results

### 3.1. Primary Outcome: Change in Infants’ HF Power during the Second Session

MT was associated with greater increase in HF compared to SSC during the second session. In the entire sample, during the second session, HF power of the infants increased by 4.35 ms^2^/Hz (SE = 0.96, *p* < 0.001), where in the MT condition it was further increased by 5.19 ms^2^/Hz (SE = 1.27, *p* < 0.001; Table 2, Figure 1, panel a). During the entire session, HF values were 2.55 ms^2^/Hz (SE = 0.94, *p* = 0.027) higher in the MT group compared to SSC.

During MT, the observed mean values of HF increased by 9.8 ms^2^/Hz from 9.86 in the first part of session to 19.66 in the last part, compared to a smaller increase of 4.24 ms^2^/Hz from 8.45 to 12.69 during SSC, with between-group effect size during the last part of the session of d = 1.46 (95% CI 0.87 to 2.05; Table 3).

In an additional analysis of each parent separately, the interaction between session part and treatment type was significant only in mothers (Appendix A).

Examination of covariates: Infants’ sex, GA, age at study entry, birthweight, and neonatal medical index grade were entered into the LME. None were predictive of HF change or influenced the demonstrated effect of MT (Appendix A).

### 3.2. Secondary Outcomes

#### 3.2.1. LF Power and LF/HF Ratio in the Second Session

Results are shown in Table 4, and Figure 1, panels b and c. Infants’ LF power decreased during the second session (B = −8.38, SE = 0.70, *p* < 0.001); was generally lower with fathers than with mothers (B = −4.05, SE = 1.13, *p* < 0.001); and lower during MT than with SSC (B = −4.00, SE = 0.81, *p* = 0.001). However, the mean decrease in LF over the session was smaller during MT than SSC by 2.4 ms^2^/Hz (SE = 0.93, *p* = 0.011), with effect size of d = −0.635 (95% CI: −1.16 to −0.100; Table 3). The interaction of parent and treatment type was also significant, with a difference of 3.78 ms^2^/Hz (SE = 1.34, *p* = 0.006).

Infants’ LF/HF ratio was also generally decreased during the second session by −1.57 ms^2^/Hz (SE = 0.14, *p* < 0.001). LF/HF ratio was lower with fathers than with mothers (B = −0.55, SE = 0.22, *p* = 0.015); lower with MT than with SSC (B = −0.81, SE = 0.14, *p* < 0.001); and decreased less during MT than during SSC (B = 0.38, SE = 0.18, *p* = 0.039). The observed mean values decreased from 1.71 to 0.53 in MT, compared to a larger decrease from 2.52 to 0.90 in SSC, with effect size of *d* = −1.37 (95% CI −1.94 to −0.78; Table 3).

#### 3.2.2. HRV Parameters across All Three Sessions

See Table 5 and Figure 2, panels a–c. The HF power was higher in the second session compared to the first session (B = 1.16, SE = 0.66, *p* = 0.078), and further increased in the third session by 0.83 (SE = 0.95) in the MT group only. No significant differences were demonstrated.

The LF power decreased from the first to the second session by −3.24 ms^2^/Hz (SE = 0.82, *p* < 0.001), but increased in the third session by 3.73 ms^2^/Hz (SE = 0.89, *p* < 0.001). LF was lower in sessions with fathers than with mothers (B = −6.62, SE = 0.98, *p* < 0.001), and significantly lower in MT in the third session, compared to SSC alone (B = −2.74, SE = 1.20, *p* = 0.023). Finally, LF was higher in sessions with fathers compared to mothers within the MT group (B = 3.00, SE = 1.27, *p* = 0.020).

Similarly, LF/HF ratio decreased from the first to the second session (B = −0.38, SE = 0.10, *p* < 0.001) and increased in the third session (B = 0.30, SE = 0.11, *p* = 0.008). LF/HF ratio was lower with fathers than with mothers (B = −0.76, SE = 0.12, *p* < 0.001), and was significantly lower in the third session in MT (B = −0.42, SE = 0.15, *p* = 0.006). Finally, LF/HF ratio was higher in MT sessions with fathers (B = 0.46, SE = 0.16, *p* = 0.004). The observed values of all HRV indices across all sessions are provided in Appendix A.

### 3.3. Parental Outcomes

Parent-to-infant attachment scores across the three sessions were smaller by −3.26 (SE = 1.75) points in the MT group compared to SSC (*p* = 0.100). No statistical significance was demonstrated (Table 6). The mean scores in both groups ranged from 72.10 (SD = 14.45, *p* = 0.741) to 81.67 (SD = 5.77, *p* = 0.031) (Appendix A), retaining a moderate–high level of parent-to-infant attachment across all measurements.

Parental state-anxiety levels decreased by 4.72 (SE = 1.25) points from pre- to post-tests in the entire sample (*p* < 0.001) (Table 6). The MT group showed a nonsignificant tendency for greater decrease in anxiety levels by 3.59 points (*p* = 0.106) across the three sessions, with a further decrease of 2.74 points from pre- to post-measurements, as compared to SSC (*p* = 0.218; Figure 3). In both groups, the mean STAI scores at all assessments ranged from 22.85 (SD = 3.17, *p* = 0.686) to 35.74 (SD = 13.62, *p* = 0.184), representing normal anxiety levels (Appendix A).

Adverse events: In the SSC group, one infant’s participation had to be paused for a week due to recurring need of ventilation. No serious adverse events occurred.

## 4. Discussion

Music therapy added to SSC resulted in improved ANS stability in preterm infants, as was indicated by the significant increase in HF power of their HRV, and higher values of HF during the entire session, compared to SSC alone. These findings support our assumption that during family-centered MT, infants’ HF power would reach around 20 ms^2^/Hz, thus further extending previous findings regarding the effectiveness of maternal singing during SSC [17]. HF power is an indicator of a relaxed state of the ANS [22,38,39], representing improvement in autonomic regulation, which strongly impacts preterm-infants’ processes of recovery and maturation. Elevated neonatal stress exposure may severely harm their ANS development and function [40,41,42], and negatively affect neurodevelopment and social–emotional participation [6,43]. Alternatively, reduced stress improves autonomic stability [5], and improved autonomic stability increases the infant’s ability to handle the external stressors of their environment, thus improving developmental outcomes [43]. Accordingly, an immediate stress-reduction intervention, facilitated by parents, is highly valuable for their developmental outcomes [1]. No significant differences between mothers and fathers were indicated, possibly due to the much smaller sample size of fathers.

During both parts of session 2, LF and LF/HF ratio values were significantly lower in the MT group compared to SSC alone. In both groups, these values decreased significantly by the end of session, suggesting that the effect of time spent in SSC was stronger than treatment type. This pattern of improved stabilization during the session was also seen in the HF analysis, and may be related to the preceding effect of SSC on improvement of HRV parameters [8,44]. However, the smaller change in LF activity in the MT group and the respective bigger change in LF activity in the SSC group cohere with the higher activity of HF that was demonstrated in the MT group and the lower HF activity that was demonstrates in the SSC group. Furthermore, the interpretation of LF power and thus LF/HF ratio may not be purely indexed as sympathetic activity, in the same way that HF was strongly correlated to parasympathetic activity in the ANS; rather, as described by Billman [45] it may represent “a complex and not easily discernible mix of sympathetic, parasympathetic, and other unidentified factors with parasympathetic factors accounting for the largest portion of the variability in this frequency range. As a consequence, the physiological basis for LF/HF is difficult to discern.” Nevertheless, at the three-month follow-up, the effect of MT was clear. HF power had increased from the second to third session only in the MT group, and LF and LF/HF values were significantly lower in the MT group. In both groups, LF and LF/HF ratio values were higher, compared to the NICU sessions. This finding is congruent with a previous study that stated that the ANS in preterm infants stabilizes during the first few months [46].

Parental STAI evaluations showed a trend toward a larger decrease in the MT group. However, parents’ mean STAI scores were below the clinical cutoff of 39–40 at all assessments [47], which may have impeded our ability to find effects on anxiety. Similarly, the MPAS did not yield any significant group differences. This questionnaire may not have been suited to the short-term intervention that was offered.

## 5. Strengths and Limitations

The relatively small number of sessions may have influenced our ability to test therapeutic processes [48], as seen in the lack of change in parental outcomes in both groups. Validated clinical cut-offs were not available for all measured outcomes, making clinical interpretation difficult. Outcome assessors were not blinded; however, the primary outcome was based on objective measurements. Furthermore, HRV data were preprocessed by an external researcher not involved in the study. Therefore, the effect on the primary outcome appears to be robust. The broad inclusion criteria, use of the principles of the worldwide RBL model [25], and the setting of the participating NICU, including shared-space accommodations and international family-centered care principles [49], further contribute to the generalizability of the study results.

The present study provided only immediate and short-term results of HRV indexes at around 34 weeks postnatal age during NICU admission period and at three-months corrected age. Further studies evaluating long-term effects of MT on preterm-infants’ ANS stabilization and neurobehavioral development are warranted. As HRV power correlates with increased age and ANS maturation [22], future studies may further clarify this process using stratified analysis on preterm infants of different gestational ages and days, to study the correlation between different gestational ages and days and HRV indexes. Additionally, to deepen the understanding of the effects of MT on ANS stabilization in newborns, further studies comparing preterm and full-term infants are warranted.

Finally, the preceding beneficial effects of SSC alone on ANS stability in preterm infants [8,38,39,44,50] may have reduced the observed effects of MT. However, the clinical experience in this NIDCAP certified NICU, as well as the results of the present study, suggest that implementing MT during SSC successfully involves all relevant aspects of protective, multi-sensory family-centered care. Furthermore, SSC position is a clear, protocolized condition, which enabled the comparison between groups much more clearly (in contrast to having the MT intervention during other holding positions).

## 6. Conclusions

Increased HRV, and specifically HF-HRV power, indicate preterm infants’ ANS stability and ability to modulate stress, where decreased HRV can be correlated with pathological conditions [51]. According to the neurovisceral integration model, HRV analysis provides a unique insight to brain activity and the heart–brain connection, as it indexes the cardiac vagal tone, and may reflect the functionality of neural networks that are implicated in regulation of physiological, affective, and cognitive processes [52]. This study demonstrated that a family-centered MT intervention during SSC led to an increased HF-HRV power. Accordingly, combined MT and SSC can be suggested as an important intervention in the routine care of preterm infants, by contributing to their ANS stability.

## Figures and Tables

**Figure 1 children-08-01077-f001:**
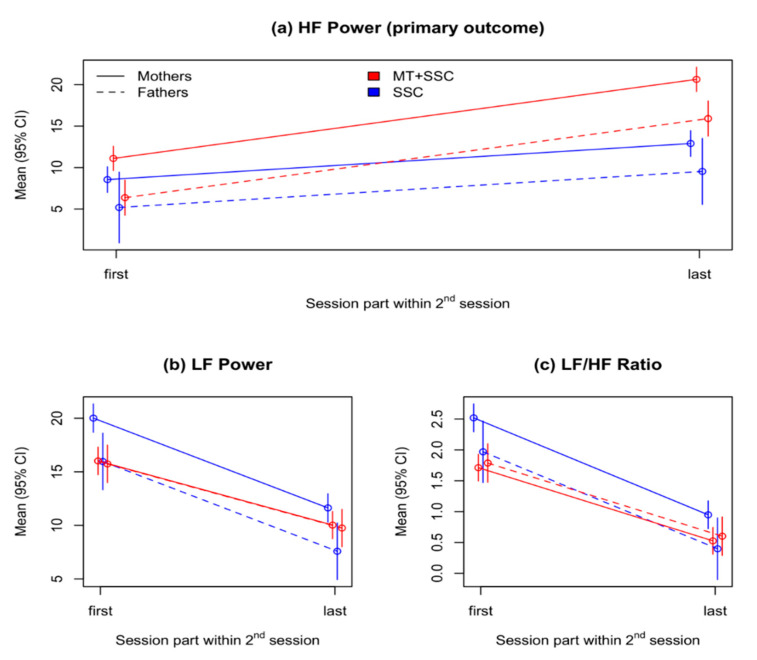
HRV parameters during the second session. (**a**): Primary outcome—Change in HF parameters during the second session; (**b**): Change in LF power; (**c**): Change in LF/HF ratio.

**Figure 2 children-08-01077-f002:**
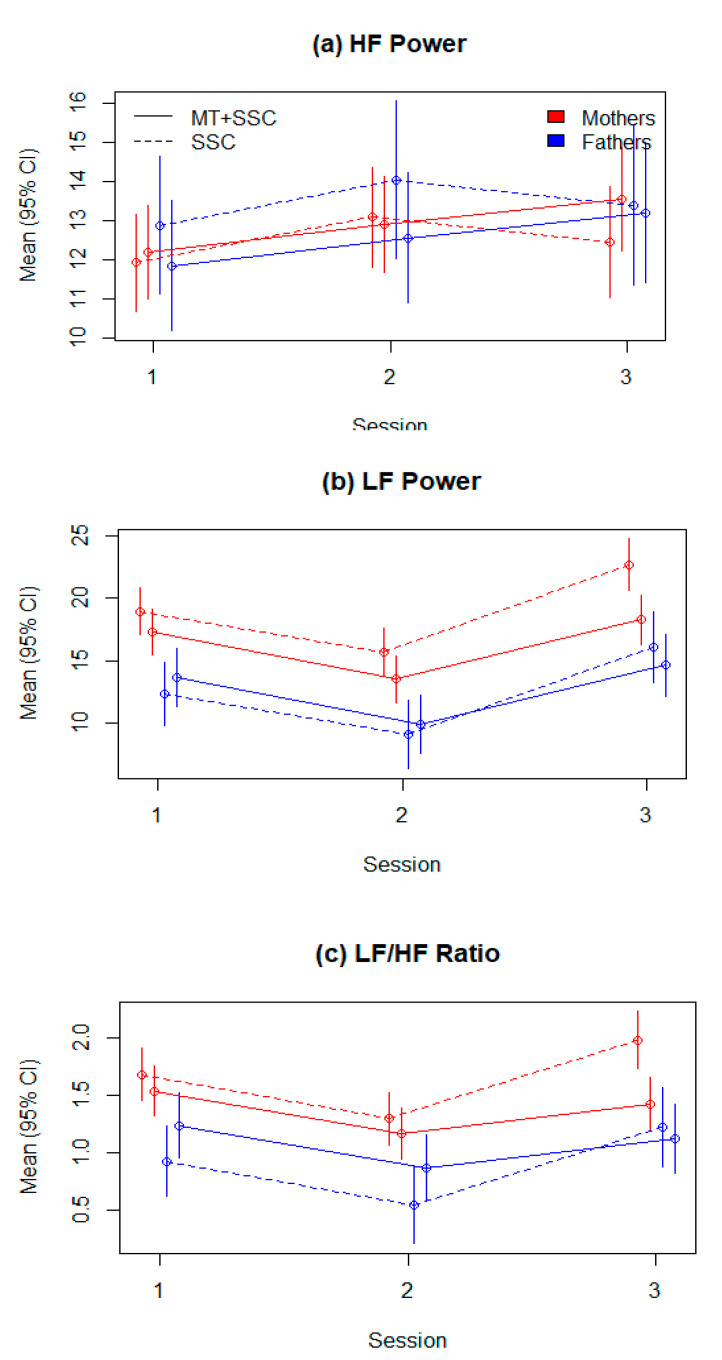
HRV parameters across all three sessions. (**a**): HF; (**b**): LF; (**c**): LF/HF ratio.

**Figure 3 children-08-01077-f003:**
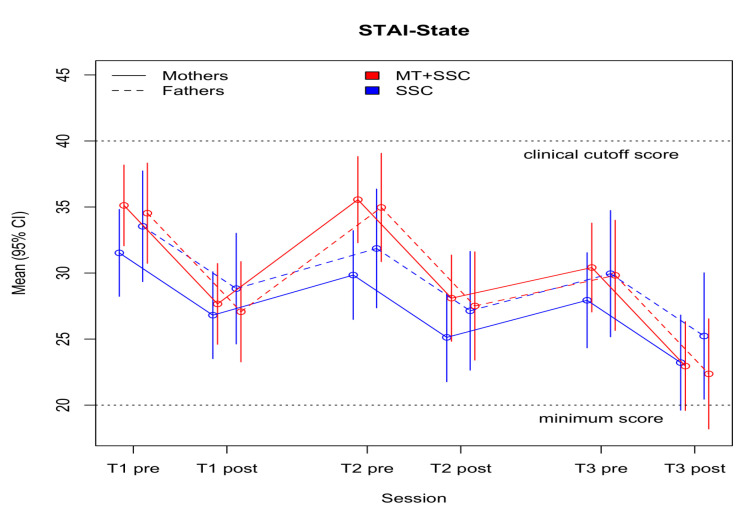
Parental state-anxiety measures across all sessions.

**Table 1 children-08-01077-t001:** Baseline characteristics.

Characteristics	MT + SSC	SSC	*p*-Value ^a^
**Neonatal**	*N* = 37	*N* = 31	
Gestational age, wk ^b^	30.56 ± 2.66	31.06 ± 2.92	0.47
Birth weight, g ^b^	1474.86 ± 494.00	1492.84 ± 460.10	0.88
Age at study entry, d ^b^	29.08 ± 24.44	24.19 ± 16.93	0.34
Age at study entry, wk ^b^	34.74 ± 1.92	34.45 ± 1.70	0.54
Male sex ^c^	15 (40%)	18 (58%)	0.15
Neonatal medical index grade 1/2/3 ^c^	11(30%)/20 (54%)/ 4 (12%)	12 (39%)/13 (42%)/ 6 (19%)	0.33
AGA/SGA ^c^ Ethnic Origin ^c^	31 (84%)/ 5 (13%)	24 (77%)/ 7 (23%)	0.43
Jewish/Arab	35(95%)/2(5%)	28(90%)/3(10%)	0.50
**Parental**	*N* = 42	*N* = 37	
Female sex ^c^	30 (71%)	27 (73%)	
Participation of one parent/ Both parents ^c^	28 (76%)/9 (24%)	26 (84%)/5 (16%)	0.40

Abbreviations: MT, music therapy; SSC, skin-to-skin care; AGA/SGA—appropriate/ small for gestational age; ^a^ equal variances not assumed; independent-sample *t*-tests were used for continuous variables and Pearson Chi-squared tests for categorical variables; ^b^ Mean ± SD; ^c^
*n* (%).

**Table 2 children-08-01077-t002:** Linear mixed effects (LME) model of primary outcome—high-frequency (HF) change in the second session.

HF Power Change in Second Session	B (SE)	*p*-Value
No. of observations: 124	ms^2^/Hz SD	
Intercept ^a^	8.55 (0.69)	<0.001
Session part: last	4.35 (0.96)	<0.001
Parent: father	−3.36 (1.83)	0.068
Treatment: MT + SSC	2.55 (0.94)	0.027
Session part: last X treatment: MT + SSC	5.19 (1.27)	<0.001
Parent: father X treatment: MT + SSC	−1.36 (2.06)	0.510

Abbreviations: MT, music therapy; SSC, skin-to-skin care. B, beta coefficient; SE, standard error; ^a^ Intercept is the predicted value for mothers in the first part of the second SSC session.

**Table 3 children-08-01077-t003:** Observed values of HRV parameters in the second session.

HRV Variable	MT + SSC	SSC	*p*-Value ^a^	Effect Size (95% CI)
	ms^2^/Hz (SD)	ms^2^/Hz (SD)		
Mean HF, first part	9.86 (2.44)	8.45 (2.02)	0.020	0.623 (0.084, 1.156)
Mean HF, last part	19.66 (6.26)	12.69 (1.23)	<0.001	1.461 (0.866, 2.046)
Mean LF, first part	15.88 (2.52)	19.88 (2.82)	<0.001	−1.501 (−2.089, −0.903)
Mean LF, last part	9.97 (2.10)	11.41 (2.45)	0.023	−0.635 (−1.165, −0.100)
Mean LF/HF ratio, first part	1.71 (0.57)	2.52 (0.66)	<0.001	−1.320 (−1.893, −0.736)
Mean LF/HF ratio, last part	0.53 (0.29)	0.90 (0.27)	<0.001	−1.368 (−1.945, −0.781)

Abbreviations: HF, high frequency; LF, low frequency; MT, music therapy; SSC, skin-to-skin care; ^a^ equal variances not assumed; *t*-tests for continuous variables; following intention-to-treat principle.

**Table 4 children-08-01077-t004:** Linear mixed effects models of secondary outcomes in the second session.

LF Change in Second Session	B (SE)	*p*-Value
No. of observations: 127		
Intercept ^a^	20.01 (0.59)	<0.001
Session part: last	−8.38 (0.70)	<0.001
Parent: father	−4.05 (1.13)	<0.001
Treatment: MT + SSC	−4.00 (0.81)	0.001
Session part: last X treatment: MT + SSC	2.40 (0.93)	0.011
Parent: father X treatment: MT + SSC	3.78 (1.34)	0.006
**LF/HF Ratio Change in Second Session**		
No. of observations: 126		
Intercept	2.52 (0.10)	<0.001
Session part: last	−1.57 (0.14)	<0.001
Parent: father	−0.55 (0.22)	0.015
Treatment: MT + SSC	−0.81 (0.14)	<0.001
Session part: last X treatment: MT + SSC	0.38 (0.18)	0.039
Parent: father X treatment: MT + SSC	0.62 (0.26)	0.019

Abbreviations: MT, music therapy; SSC, skin-to-skin care; B, beta coefficient; SE, standard error; ^a^ Intercept is the predicted value for mothers in the first part of the second SSC session.

**Table 5 children-08-01077-t005:** Linear mixed effects models of secondary HRV outcomes across all three sessions.

HF Power across Sessions	B (SE)	*p*-Value
No. of observations: 199		
Intercept ^a^	11.93 (0.55)	<0.001
Second session	1.16 (0.66)	0.078
Third session	0.52 (0.71)	0.462
Parent: fathers	0.95 (0.76)	0.212
Treatment: MT + SSC	0.27 (0.75)	0.725
Second session X treatment: MT + SSC	−0.45 (0.89)	0.608
Third session X treatment: MT + SSC	0.83 (0.95)	0.385
Parent: fathers X treatment: MT + SSC	−1.30 (0.99)	0.194
**LF Power across Sessions**		
No. of observations: 199		
Intercept	18.92 (0.82)	<0.001
Second session	−3.24 (0.82)	<0.001
Third session	3.73 (0.89)	<0.001
Parent: fathers	−6.62 (0.98)	<0.001
Treatment: MT + SSC	−1.68 (1.13)	0.177
Second session X treatment: MT + SSC	−0.51 (1.11)	0.646
Third session X treatment: MT + SSC	−2.74 (1.20)	0.023
Parent: fathers X treatment: MT + SSC	3.00 (1.27)	0.020
**LF/HF Power across Sessions**		
No. of observations: 198		
Intercept	1.68 (0.10)	<0.001
Second session	−0.38 (0.10)	<0.001
Third session	0.30 (0.11)	0.008
Parent: fathers	−0.76 (0.12)	<0.001
Treatment: MT + SSC	−0.14 (0.14)	0.325
Second session X treatment: MT + SSC	0.02 (0.14)	0.897
Third session X treatment: MT + SSC	−0.42 (0.15)	0.006
Parent: fathers X treatment: MT + SSC	0.46 (0.16)	0.004

Abbreviations: MT, music therapy; SSC, skin-to-skin care. ^a^ Intercept is the predicted value for mothers in the first part of the second SSC session; B, beta coefficient; SE, standard error.

**Table 6 children-08-01077-t006:** Linear mixed effects models of parental outcomes across sessions.

Maternal Postnatal Attachment Scale	B (SE)	*p*-Value
No. of observations: 166		
Intercept ^a^	81.69 (1.31)	<0.001
One month assessment	0.14 (2.00)	0.942
Three-month follow-up	−1.88 (1.82)	0.303
Parent: fathers	−1.37 (2.10)	0.517
Treatment: MT + SSC	−3.26 (1.75)	0.100
One month assessment X Treatment: MT + SSC	−1.11(2.97)	0.709
Three months X treatment: MT + SSC	2.98 (2.41)	0.219
Parent: father X treatment: MT + SSC	−1.40 (2.71)	0.606
**State–Trait Anxiety Inventory**		
No. of observations: 400		
Intercept	31.53 (1.46)	<0.001
Second session	−1.68 (1.46)	0.252
Third session	−3.59 (1.59)	0.024
Session part: post	−4.72 (1.25)	<0.001
Parent: fathers	2.01 (1.61)	0.213
Treatment: MT + SSC	3.59 (1.97)	0.106
Second session X treatment: MT + SSC	2.11 (1.98)	0.286
Third session X treatment: MT + SSC	−1.12 (2.10)	0.596
Session part: post X treatment: MT + SSC	−2.74 (1.67)	0.102
Parent: father X treatment: MT + SSC	−2.60 (2.11)	0.218

Abbreviations: MT, music therapy; SSC, skin-to-skin care. ^a^ Intercept is the predicted value for mothers in the pre-test, first session in SSC group; B, beta coefficient; SE, standard error.

## Data Availability

The data presented in this study are available on request from the corresponding author.

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
