# Peer review of "Effects of Live Music Therapy on Autonomic Stability in Preterm Infants: A Cluster-Randomized Controlled Trial"

_children, 2021, doi:10.3390/children8111077_

Round 1
Reviewer 1 Report
I think this was a well done study but I have some suggestions. First, it is suggested to establish term infants as the control group. The significance is to judge whether there is a difference in autonomic nerve stability between preterm infants and normal term infants. SSC practice is considered standard care in the participating NICU. So there is no MT group which receive music therapy only. The effects of MT on preterm-infants' ANS stabilization and neurobehavioral development was unclear. Conclusion should be changed to “MT + SSC can be suggested as an important intervention in the routine care of preterm infants, by contributing to their ANS stability.” Whether there is a mixed effect in the combination of MT + SSC group are warranted. Second, the physiological mechanisms affecting HRV include respiration, baroreflex and thermoregulation. Physiological immaturity is the main reason for the decrease of HRV in preterm infants. Neonatal pathological phenomena can affect HRV. Apnea, hyaline membrane disease and intracranial hemorrhage can reduce HRV. The exclusion criteria should be more specific. Third, the detection of HRV provides a method to measure the function of sympathetic and parasympathetic nerves. It is suggested to conduct stratified analysis on preterm infants of different gestational ages and days to study the correlation between different gestational ages and days and HRV indexes.
Author Response
Response to reviewer 1 comments:
Point 1 : I think this was a well done study but I have some suggestions. First, it is suggested to establish term infants as the control group. The significance is to judge whether there is a difference in autonomic nerve stability between preterm infants and normal term infants.
Response to point 1: Thank you so much for the informed suggestion to have a control group of full-term infants. Our study has focused on preterm infants' autonomic nervous system stability (ANS) since the effects of stress exposure during the neonatal intensive care period may have a crucial negative impact for their developing autonomic system, and therefore for their developmental outcomes. The main aim was to test the effects of music therapy in preterm infants, hence a control group with matching baseline characteristics of gestational age and medical condition was deemed appropriate.
In addition, we agree that it is important to “benchmark” the values against those of some comparison group. One such comparison is provided in the manuscript in the power calculation section (page 5, lines 204-211) The power calculation was based on a previous study (Arnon et al., 2014), which found mean values of 16.8 ms2/Hz for HF power during combined maternal singing and SSC, as compared to 10.5 ms2/Hz during SSC alone (effect size d=1.44). Accordingly, we hypothesized that in the current study HF values may increase more, up to 20 ms2/Hz, due to the family-centered approach. Finally, this hypothesize was supported by the current trial's results of HF values during the second session (page 11, lines 323-326).
Nevertheless, we have added the suggestion to have another trial comparing the effects of MT on ANS stability between preterm and term infants to the concluding discussion as a recommended future examination which can deepen the understanding of the effects of music therapy on ANS stabilization in newborns (page 12, lines 380-381)
" Additionally, to deepen the understanding of the effects of MT on ANS stabilization in newborns, further studies comparing preterm and full-term infants are warranted. “
Point 2: SSC practice is considered standard care in the participating NICU. So there is no MT group which receive music therapy only.
Response to point 2: The decision to combine music therapy and SSC was based on recent evidence suggesting that provision of live music or the parental voice may consider both sensory regulation precautions and enhancement of family centered care (as described in the introduction section, page 2, lines 48-56). Furthermore, SSC position is a clear, protocolized condition, which enabled the comparison between groups much clear (in contrast to having the music therapy intervention during other holding positions). In response to this comment, we have added this explanation to the discussion of the trial's strengths and limitation section (Page 12, lines 363-388):
"Finally, the preceding beneficial effects of SSC alone on ANS stability in preterm-infants [8,38,39,44,50] may have reduced the observed effects of MT. However, the clinical experience in this NIDCAP certified NICU, as well as the results of the present study, suggest that implementing MT during SSC successfully involves all relevant aspects of protective, multi-sensory family-centered care. Furthermore, SSC position is a clear, protocolized condition, which enabled the comparison between groups much clear (in contrast to having the MT intervention during other holding positions)."
Point 3: The effects of MT on preterm-infants' ANS stabilization and neurobehavioral development was unclear. Conclusion should be changed to “MT + SSC can be suggested as an important intervention in the routine care of preterm infants, by contributing to their ANS stability.” Whether there is a mixed effect in the combination of MT + SSC group are warranted.
Response to point 3: Thank you for your advice. The conclusion was changed accordingly (page 12, lines 397-399).
Point 4: Second, the physiological mechanisms affecting HRV include respiration, baroreflex and thermoregulation. Physiological immaturity is the main reason for the decrease of HRV in preterm infants. Neonatal pathological phenomena can affect HRV. Apnea, hyaline membrane disease and intracranial hemorrhage can reduce HRV. The exclusion criteria should be more specific.
Response to point 4: Thank for your clarification. We have now detailed the exclusion criteria further to include the varied severe medical conditions which may affect HRV, which we had considered, but had not described previously: (page 3, lines 101-107):
Infants were excluded due to medications acting on the central nervous system, any respiratory distress with oxygen support, intraventricular hemorrhage stage ≥ 3, periventricular leukomalacia, active apnea episodes which require medical intervention, or an estimated hospitalization of less than ten days. Additionally, pre study data was collected from medical charts and included bronchopulmonary dysplasia, necrotizing enterocolitis, respiratory distress syndrome, retinopathy of prematurity and sepsis episodes.
Point 5: Third, the detection of HRV provides a method to measure the function of sympathetic and parasympathetic nerves. It is suggested to conduct stratified analysis on preterm infants of different gestational ages and days to study the correlation between different gestational ages and days and HRV indexes.
Response to point 5: Thank you for your suggestion. Since we aimed to compare the effects of MT on ANS stability in preterm infants, we thought it is important to compare groups with similar baseline characteristics. We did examine gestational age as a predictor, as part of the covariates examination, and did not find any effects of gestational age for the primary outcomes of HF power (page 7, lines 254-256; Table S2). Additionally, in the discussion section we mention that the follow-up session, which was conducted at 3-months corrected age has showed increased HRV indexes in both groups. This finding supports a previous study that stated that "the ANS in preterm infants stabilizes during the first few months" (De Rogalski et al., 2007) (page 12, line 354-356). Due to our short-term design, we could provide examination of HRV indexes only at 34 and 35 weeks postnatal age during NICU admission period and at 3-months corrected age. We admitted the limitations of the current analysis and added the suggestion to conduct a correlational analysis of different gestational ages and days and HRV indexes to the discussion section (page 12, lines 373-379):
"The present study provided only immediate and short-term results of HRV indexes at around 34 weeks postnatal age during NICU admission period and at 3-months corrected age. Further studies evaluating long-term effects of MT on preterm-infants' ANS stabilization and neurobehavioral development are warranted. As HRV power correlates with increased age and ANS maturation [Longin et al., 2006], future studies may further clarify this process using stratified analysis on preterm infants of different gestational ages and days, to study the correlation between different gestational ages and days and HRV indexes. "

Reviewer 2 Report
In this study authors aimed to evaluate if the combined MT and SSC will enhance ANS regulation, will decrease parental stress and increase parent-to-infant attachment.
It is a well-conducted study, with a small number of participants. I have some comments for authors:
Please stained the aim of the study in abstract
Lines 92-94 give more information about the confirmed hearing based on otoacoustic emissions test. The mean Gestational age, (30.56 ± 2.66 and 31.06 ± 2.92 .47 weeks respectively for each group); how and when the otoacoustic emissions test were performed.
Lines 230-233: this section is referred to secondary outcome.
Lines 261-265 and 281-286, these results should explained and commented in discussion section.
Conclusions section: : is too strong to be supported based on the results of the study.
Author Response
Response to reviewer 2 comments:
Point 1: In this study authors aimed to evaluate if the combined MT and SSC will enhance ANS regulation, will decrease parental stress and increase parent-to-infant attachment.
It is a well-conducted study, with a small number of participants. I have some comments for authors:
Please stained the aim of the study in abstract
Response to point 1: Thank you for your advice. We have stressed the aim of the study in the abstract: page 1, lines 17-18: "Accordingly, we aimed to compare the effects of combined MT and SSC and SSC alone on ANS stabilization in preterm infants"
Point 2: Lines 92-94 give more information about the confirmed hearing based on otoacoustic emissions test. The mean Gestational age, (30.56 ± 2.66 and 31.06 ± 2.92 .47 weeks respectively for each group); how and when the otoacoustic emissions test were performed.
Response to point 2: Thank you for the detailed guidance. The required information was added to the manuscript. See page 3, lines 96-99: " All clinically stable preterm infants with gestational age (GA) < 36 weeks and confirmed hearing based on otoacoustic emissions test were eligible. The hearing test was conducted prior to study entry by a specialized assessor at the NICU environment. ".
Infants' mean gestational age was added to the written information (in addition to Table 1: Baseline characteristics). Page 3, lines 114-116.
Point 3: Lines 230-233: this section is referred to secondary outcome.
Response to point 3: The subtitles were better defined: (lines 259-258)
"3.2. Secondary outcomes
3.2.1. LF power and LF/HF ratio in the second session.
We hope this is better clarified now.
Point 4: Lines 261-265 and 281-286, these results should explained and commented in discussion section.
Response to point 4: Lines 340-353 discuss the secondary HRV outcomes in the second session, i.e., LF and LF/HF ratio:
"During both parts of session 2, LF and LF/HF ratio values were significantly lower in the MT group compared to SSC alone. In both groups, these values decreased significantly by the end of session, suggesting that the effect of time spent in SSC was stronger than treatment type. This pattern of improved stabilization during the session was also seen in the HF analysis, and may be related to the preceding effect of SSC on improvement of HRV parameters [8,45]. However, the smaller change in LF activity in the MT group and the respective bigger change of LF activity in the SSC group is adequate to the higher activity of HF that was demonstrated in the MT group. Furthermore, the interpretation of LF power and thus LF/HF ratio may not be purely indexed as sympathetic activity, in the same way that HF was strongly correlated to parasympathetic activity in the ANS; rather, as described by Billman [46] it may represent "a complex and not easily discernible mix of sympathetic, parasympathetic, and other unidentified factors with parasympathetic factors accounting for the largest portion of the variability in this frequency range. As a consequence, the physiological basis for LF/HF is difficult to discern"
The secondary outcomes of HRV indices across the 3 sessions are discussed through lines 353-356:
"HF power had increased from the second to third session only in the MT group, and LF and LF/HF values were significantly lower in the MT group. In both groups, LF and LF/HF ratio values were higher, compared to the NICU sessions. This finding is congruent with a previous study that stated that the ANS in preterm infants stabilizes during the first few months".
Point 5: Conclusions section: is too strong to be supported based on the results of the study.
Response to point 5: Incorporating the comments from both reviewers we have re-defined the study conclusions and stressed the combination of music therapy and SSC as a recommended beneficial intervention:
Lines 397-399:
" Accordingly, combined MT and SSC can be suggested as an important intervention in the routine care of preterm infants, by contributing to their ANS stability."
Round 2
Reviewer 1 Report
Thank you for your reply.
Reviewer 2 Report
In this revised version, the authors have significantly improved the quality of the manuscript.